# Development of a Cavitation Generator Mimicking Pistol Shrimp

**DOI:** 10.3390/biomimetics9010047

**Published:** 2024-01-12

**Authors:** Hitoshi Soyama, Mayu Tanaka, Takashi Takiguchi, Matsuo Yamamoto

**Affiliations:** 1Department of Finemechanics, Tohoku University, Sendai 980-8579, Japan; 2School of Dentistry, Showa University, Tokyo 145-8515, Japan

**Keywords:** cavitation, cavitation generator, pistol shrimp, pulsed laser, pulsed water jet, piezo actuator

## Abstract

Pistol shrimp generate cavitation bubbles. Cavitation impacts due to bubble collapses are harmful phenomena, as they cause severe damage to hydraulic machinery such as pumps and valves. However, cavitation impacts can be utilized for mechanical surface treatment to improve the fatigue strength of metallic materials, which is called “cavitation peening”. Through conventional cavitation peening, cavitation is generated by a submerged water jet, i.e., a cavitating jet or a pulsed laser. The fatigue strength of magnesium alloy when treated by the pulsed laser is larger than that of the jet. In order to drastically increase the processing efficiency of cavitation peening, the mechanism of pistol shrimp (specifically when used to create a cavitation bubble), i.e., *Alpheus randalli*, was quantitatively investigated. It was found that a pulsed water jet generates a cavitation bubble when a shrimp snaps its claws. Furthermore, two types of cavitation generators were developed, namely, one that uses a pulsed laser and one that uses a piezo actuator, and this was achieved by mimicking a pistol shrimp. The generation of cavitation bubbles was demonstrated by using both types of cavitation generators: the pulsed laser and the piezo actuator.

## 1. Introduction

Cavitation is a phase change phenomenon, wherein a change from a liquid phase to a gas phase is achieved via a decrease in pressure due to an increase in flow velocity [1]. The phenomenon was discovered as occurring on screw propellers in the 1890s [2]. As the impacts induced by cavitation collapse cause severe damage in hydraulic machinery, such as in pumps, valves, etc., cavitation is a harmful phenomenon. In order to evaluate the cavitation resistance of materials, cavitation erosion tests were standardized in ASTM International as ASTM G32 [3] and G134 [4]. As per ASTM G32 and G134, cavitation is generated by a vibratory horn at an ultrasonic band, i.e., ultrasonic cavitation, and a submerged high-speed water jet with cavitation, i.e., a cavitating jet. On the other hand, cavitation impacts can be utilized for mechanical surface treatment to improve fatigue strength, which is called cavitation peening [5]. It is worthwhile to develop a novel cavitation generator for mechanical surface treatment.

In conventional mechanical surface treatments, shot peening (in which shots are impinged to target materials), is used [6,7,8,9,10,11,12,13]. Shot peening can improve the fatigue properties of metallic materials [14,15,16,17,18]; however, the surface roughness is increased by solid collisions [19,20]. When the fatigue strength of peened stainless steel was compared, the fatigue strength of cavitation peening was found to be better than that of shot peening [21]. The residual stress and the relaxation in the compressive residual stress introduced by peening were discussed in the fatigue life of peened stainless steel [22]. It was reported that the relaxation of cavitation peening was less than that of shot peening [23]. As the dislocation density of cavitation-peened stainless steel was also smaller than that of shot peening at equivalent compressive residual stress conditions, it was found that this is one of reasons why the relaxation of cavitation peening is less than that of shot peening. It was also reported by using tensile testing that the yield stress of stainless steel increased from 6% to 8% with increasing high strain rate [24,25,26,27]. Cavitation peening is a kind of shockwave process that occurs at bubble collapse; thus, the strain rate of cavitation peening is larger than that of shot peening. Thus, cavitation peening can treat metallic materials with a small increase in dislocation density, thereby resulting in a lesser relaxation in the compressive residual stresses than shot peening. Then, the fatigue strength of stainless steel treated by cavitation peening was improved from 279 MPa of a non-peened one to 348 MPa, which was better than shot peening, i.e., 325 MPa [21]. It was also reported that the improvement in the fatigue strength of gears treated by cavitation peening was 24%, and in those treated by shot peening it was 12% compared with non-peened gears. As such, it is worthwhile to develop cavitation peening in order to improve the fatigue strength of metallic materials.

In conventional cavitation peening, cavitation is produced by a cavitating jet (see Figure 1a) [28]. When the jet impinges to the target, a cloud cavitation is generated near a nozzle and becomes a ring vortex cavitation. Then, a part of the ring vortex cavitation collapses as a longitudinal vortex cavitation generates impacts [28]. Specifically, it is vortex cavitations [29,30], which are similar to horseshoe cavitations [31,32], that generate severe impacts. Unfortunately, it is 1/1000–1/100 the value of vortex cavitations that generate intense cavitation [5], and the mechanism for them is not clear. In another way to conduct cavitation peening, a cavitation is generated by a submerged pulsed laser, i.e., laser cavitation peening [33,34,35] (see Figure 1b). In the case of magnesium alloy, the fatigue strength of laser cavitation peening is better than cavitation peening conducted via a cavitating jet [36]. It was reported that the suitable laser pulse density of improvement of the bending fatigue strength by laser cavitation peening was 4 pulse/mm^2^ for stainless steel SUS316L [21], 5 pulse/mm^2^ for additively manufactured (AM) titanium alloy Ti6Al4V [37], and 14 pulse/mm^2^ for magnesium alloy AZ31 [36]. The Vickers hardness of these tested materials without peening were 168 ± 3 for SUS316L [21], 408 ± 11 for AM Ti6Al4V [38] and 53.0 ± 0.7 for AZ31 [36]. At laser cavitation peening, the impact is proportional to the volume of the bubble considering threshold level, and the threshold level of target material increases with Vickers hardness [34]. As AZ31 is softer than SUS316L and AM Ti6Al4V, a big impact is not required for laser cavitation peening of AZ31 at laser cavitation peening. However, the required laser pulse density for laser cavitation peening of AZ31 for the improvement of the bending fatigue strength was about three times larger than that of SUS316L and AM Ti6Al4V. Namely, laser cavitation peening of AZ31 requires a long time to treat. For example, the processing time for AZ31 at 14 pulse/mm^2^ to treat 10 cm × 10 cm area by 10 Hz, which is the maximum repetition frequency of the pulsed laser with 0.35 J of pulse energy for submerged laser peening, is 14,000 s (3 h 53 min 20 s). This is one of reasons to develop a new cavitation generator for the reduction of the recessing time. Laser cavitation peening is similar to laser peening, which is also called laser shock peening. Note that there are two types of laser peening. One is laser peening with a water film, in which the target is covered with a water film [39,40,41,42,43,44,45]. The other is submerged laser peening, in which the target is placed in water [21,46,47,48,49,50]. In both laser peening methods, the pulsed laser is irradiated to the target surface, which covers a confining medium such as water. The shock wave at laser ablation is then concentrated into the target materials by the confining medium, which then produces plastic deformation [49]. In submerged laser peening, a bubble that behaves like a cavitation bubble is generated after laser ablation, which then collapses [21,51]. The bubble induced by a pulsed laser is called laser cavitation [33]. Note that cavitation does not occur at conventional laser peening with the water film, because the water is dispersed [5].

In the case of submerged laser peening, when the amplitude of pressure in water is measured by a submerged shockwave sensor, the amplitude of laser ablation is larger than that of laser cavitation collapse [21,51]. On the other hand, the impact passing through the target has been measured by a sensor with a polyvinylidene fluoride PVDF film, and it has been found that the impact at a laser cavitation collapse is larger than that of laser ablation [21,34]. Namely, laser cavitation is applicable for cavitation peening. Unfortunately, it takes time to improve the fatigue strength by laser cavitation peening as the repetition frequency of the pulsed laser is about several dozen times per second. Although a high-repetition portable pulse laser, whose power and pulse width are 10 mJ and 1.3 ns, was developed in [52], its repetition frequency was only about 100 Hz, and 800–1600 pulses/mm^2^ were required for treatment. Even then, it still requires a long time to treat the target. In the case of laser cavitation peening, we tried to increase pulse repetition frequency from 10 Hz to 1 kHz using a normal oscillation laser without Q-switch or a fiber laser. The generation of cavitation was already demonstrated by using a normal oscillation laser without Q-switch [53], however, the bubble size was not large enough. This is why we tried to develop new cavitation generators.

Cavitation peening using an ultrasonic vibratory horn has also been proposed [54,55,56,57,58] (see Figure 1c); however, the aggressive intensity of ultrasonic cavitation is particularly sensitive to the standoff distance between the horn and target surface. For example, the cavitation intensity was reduced to approximately 20% from 0.8 mm to 1.0 mm. And, also, even though the vibratory frequency was 20 kHz, the intense cavitation impact occurred several times per second [59]. Furthermore, when considering cavitation peening with cavitating jets, submerged pulsed lasers, and ultrasonic vibratory horns, it is worthwhile to develop a novel cavitation generator, which can be used to drastically increase the processing efficiency of cavitation peening.

In nature, pistol shrimp [60,61,62,63,64,65,66,67] and mantis shrimp [68,69,70] are known to create cavitations, and the mechanical properties of their claws has also been investigated [71]. Although there was no term of “cavitation” in the references [72,73], the noise [72] and the water jet velocity [73] of the snapping pistol shrimp claw were measured. Note that there are several types of claws for pistol shrimps [74]. From the view point of mechanical devices that are bioinspired by pistol shrimp, a pulsed water jet generator was proposed [75,76], and a ring vortex cavitation around the pulsed water jet from a device was realized by mimicking the claws of pistol shrimp [77]. A bioinspired plasma generator [78] and an impulsive motion generator [79] were also developed. When the claw of a pistol shrimp is imitated, high-repetition frequencies, such as several hundred times per second, are not expected. As such, a cavitation generator for peening was developed by mimicking the essential mechanism in the generation of cavitations by pistol shrimp, which is a pulsed water jet that generates cavitation.

In the present paper, in the first part, a pistol shrimp claw, i.e., *Alpheus randalli*, was measured by a μCT, and the volume of the pulsed water jet was also estimated. Then, a cavitation bubble induced by a pistol shrimp was evaluated by comparing the noise created by laser cavitation. In the second part, a pulsed water jet is produced by mimicking a pistol shrimp, and two cavitation generators are developed using a pulsed laser and a piezo actuator. In order to generate cavitation mimicking a pistol shrimp, generation of a pulsed water jet is a key point. It was expected that a pulsed laser type could produce a high-speed pulsed water jet, and a piezo actuator type could produce a pulsed water jet, but the speed of the jet of the piezo actuator was not high enough. At the application of cavitation peening, a piezo actuator type would be useful, as it would be a compact system. This is why two cavitation generators are developed.

## 2. Materials and Methods

Figure 2 shows an examined pistol shrimp, i.e., *Alpheus randalli*. The length of its body without whiskers was about 35 mm. As shown in Figure 2, the right-hand side claw was bigger than the left-hand side, as it was responsible for creating cavitation. Figure 3 reveals the right-hand side claw of the pistol shrimp, which was its shed shell. The three-dimensional shape of the claw was measured by a dental μCT apparatus (ScanXmate-L090H, Comscan Techno, Kanagawa, Japan) and TRI-3D BONE (Ratoc CO, Tokyo, Japan) was used for image analysis. At the present observation, the field of view was 616 pixels × 828 pixels × 488 steps, and the spatial resolution was 14.64 μm/pixel, then the field of view in mm was about 9 mm × 12 mm × 7 mm. In order to estimate the bubble size produced by the pistol shrimp, the noise was detected. Figure 4 illustrates the schematics of the measurement of the noise that was generated by the pistol shrimp. The pistol shrimp stayed under the coral sands. The noise was detected by a hydrophone (Miniature Hydrophone Type 8103, Brüel & Kjær, HBK Company, Nærum, Denmark), and the signal was connected to a pre-amplifier (Charge conditioning amplifier 2692, Brüel & Kjær, HBK Company, Nærum, Denmark). Then, the signal was recorded by a digital oscilloscope (DPO3054, Tektronix, Inc., Beaverton, OR, USA). Artificial sea water (Instant ocean, Spectrum Brands, Inc., Blacksburg, VA, USA) was also used in the present experiment.

Figure 5 shows schematics of a submerged pulsed laser system, which was used to compare laser cavitation with the cavitation generated by a pistol shrimp. The used laser pulse source was a Q-switched Nd:YAG laser with a wavelength conversion (Surelite^TM^ SL I-10, Continuum^®^, Amplitude Laser Inc., San Jose, CA, USA). Considering the reported papers [21,48], 532 nm and 1064 nm were chosen. By controlling the wavelength conversion, both 532 nm and 1064 nm wavelengths could be obtained. The pulse width and the repetition frequency of the laser pulse were 6 ns and 10 Hz, respectively. Regarding a specification document, the beam diameter was 6 mm, and it was confirmed by using photographic printing paper. The maximum energy was 0.2 J at 532 nm and 0.35 J at 1064 nm. For the purposes of comparing the laser cavitation with the cavitation generated by the pistol shrimp, a 532 nm measurement was used to avoid the attenuation of the laser power by water. The pulsed laser from the laser source was reflected by mirrors, and it was expanded by a concave lens to avoid damage to the chamber. This was then focused by a convex lens of 100 mm at focal distance, and this was then finally focused by a convex lens of 15 mm at focus distance, which was subsequently placed into water.

The maximum diameter of laser cavitation *d_max_* was controlled by laser power. The *d_max_* was measured by using a high-speed video camera (VW9000, Keyence Corporation, Osaka, Japan). The full frame size was 640 pixels × 640 pixels, and the maximum frame rate at full frame was 4000 frames per second. The maximum frame rate was 230,000 frames per second and the frame size was 160 pixels × 32 pixels. Once the relationship between *t_D_* and *d_max_* was obtained, *d_max_* can be obtained from *t_D_*. Here, *t_D_* was defined by the time between the laser ablation and laser cavitation collapse. With respect to Rayleigh [80], the collapse time *t_c_* for a bubble from radius *R*_0_ to collapse is given by Equation (1).
(1)$$t_{c}={0.91468\;R}_{0}\;\sqrt{\frac{\;\rho \;}{p}}$$

As such, Equation (2) is obtained from Equation (1).
(2)$$R_{0}\;\left[mm\right]=k_{1}\;t_{c}\;[\mu s]$$

Here, *k*_1_ [mm/μs] is a proportional constant; furthermore, it is 0.0111 for a *p* = 0.1013 MPa and *ρ* = 998 kg/m^3^ of water, and it is 0.0110 for a *p* = 0.1013 MPa and *ρ* = 1003 kg/m^3^ of sea water.

In the case of *d_max_,* Equation (2) denoted Equation (3).
(3)$$d_{max}\;\left[mm\right]=k_{1}\;t_{D}\;[\mu s]$$

Specifically, *d_max_* can be obtained from *t_D_*.

Figure 6 illustrates the schematics of a cavitation generator by using the pulsed laser. The used pulsed laser was a Q-switched Nd:YAG laser with a wavelength conversion that was same as detailed in Figure 5. For the cavitation generator, the wavelength of the used pulsed laser was 1064 nm. The acrylic chamber was 10 mm in diameter and 12 mm in length, with the nozzle having a diameter of 0.5 mm. When the pulsed laser was irradiated in the acrylic chamber, a laser cavitation was generated, and the volume expansion of the laser cavitation produced a pulsed jet through the nozzle. Then, the cavitation was able to be generated around the pulsed jet.

Figure 7 shows the schematics of a cavitation generator that uses a piezo actuator. The used piezo actuator was a metal-cased type piezoelectric actuator (AHB101C801 ND0LF, TOKIN Corporation, Sendai, Japan). The actuator was operated by a high-voltage power supply (PZDR-0.15P6A, Matsusada Precision Inc., Shiga, Japan). The used signal for the power supply was generated by a transistor–transistor logic TTL signal generator (AN-PGV100, IDT Japan, Inc., Tokyo, Japan). The chamber was made by an acrylic pipe, whose inner and outer diameter were 26 mm and 30 mm, and whose length was 20 mm. In order to fill the water into the chamber, a syringe was connected to the chamber though a 0.56 mm diameter needle. The diaphragm was made by a polytetrafluoroethylene PTFE sheet of 1 mm thickness. When the piezo actuator expanded, a pulsed water jet was generated through the nozzle (whose diameter was 0.6 mm). The nozzle size was decided by considering the movement of the piezo actuator and the diameter of the chamber. As the aggressive intensity of the cavitating jet was strongly affected by the nozzle exit, the nozzle had an outlet bore that was 2 mm in diameter and 1 mm in length. The aggressive intensity of cavitation was normally evaluated by the erosion rate of materials or the arc height of the peened plate. The details are shown in the reference. The movement of the piezo actuator and the aspect of the cavitation were observed by a high-speed video camera (OS8-V3 S3, IDT Japan, Inc., Tokyo, Japan). The full frame size was 1600 pixels × 1200 pixels, and the maximum frame rate at full frame was 8000 frames per second. The maximum frame rate was 194,000 frames per second and the frame size was 1600 pixels × 16 pixels.

## 3. Results

### 3.1. Cavitation Induced by Pistol Shrimp Compared with a Pulsed Laser

Figure 8 shows the diagonal view of the opened claw that was measured by the μCT. Figure 9 reveals the aspects of the *x*, *y*, and *z* cross-sections of the closed claw that was measured by the μCT. Figure 9 shows that the movable part of the claw, which had a convex part, and the concave part of the fixed side had a shape into which the convex part was inserted. It has been reported that there are several types of pistol shrimps [74]. Some of them have a deep fossa and a large plunger [74]. In the case of *Alpheus randalli*, the convex and concave parts were relatively shallow when compared with the reported one [60]. Figure 8 reveals that the convex hull had a height and width of about 1 mm; on the other hand, Figure 9 shows that the volume of the concavo-convex mating area was much smaller than that. If the concavo-convex mating area is assumed to be a semi-ellipsoid, its height, width, and depth are estimated to be about 0.48 mm, 0.48 mm, and 0.72 mm, respectively, and its volume is estimated to be about 0.35 mm^3^.

Figure 10 shows the sound pressure *p_N_* changing with time *t* for the pistol shrimp Figure 10a and for the pulsed laser Figure 10b. To measure the sound pressure *p_N_*, the hydrophone was horizontally placed at a distance of 50 mm from the laser focus point. In Figure 10a, the sound nearest 0 μs was caused by full closure of the claw, and the noise at 300 μs was generated by the bubble collapse as similar to the reference [60]. In Figure 10b, the first peak was caused by laser ablation and the second peak was produced by the laser cavitation collapse [33]. When the difference from the noise source to the hydrophone, i.e., 50 mm for the pulsed laser and 90 mm for the pistol shrimp, was considered, the noise ratio of the pistol shrimp to the pulsed laser was found to be (90/50)^2^ ≈ 3.2. Thus, the *p_N_* of the shrimp was nearly equivalent to that of the pulsed laser. In Figure 10, the peak value was 7.7 kPa for the pistol shrimp and 25 kPa for the pulsed laser. It was then found that the noise levels of the pistol shrimp and the pulsed laser were similar. Thus, it was worthwhile to compare the noise level of the pistol shrimp with that of the pulsed laser.

In order to show the determination of the relation between *d_max_* and *t_D_* in Equation (3), Figure 11 reveals the aspect of laser cavitation, which was recorded by a high-speed video camera with the noise of Figure 10b. The bubble was developed, and it had a maximum diameter, i.e., *d_max_* at *t* = 0.147 ms, and shrunk, then collapsed at *t* = 0.295 ms, which corresponded to *t_D_*. Thus, the relation between *d_max_* and *t_D_* was obtained.

It was reported that the aggressive intensity of the cavitation bubble collapse was proportional to the volume of cavitation [28,29,81], and the volume of the bubble should be considered to compare the aggressive intensity of the cavitation induced by the pistol shrimp and the pulsed laser cavitation. As mentioned in the introduction concerning Equation (3), the developing time *t_D_* from initiation to collapse was found to be proportional to the maximum diameter *d_max_*. In order to confirm the relation between *t_D_* and *d_max_*, Figure 12 helped to reveal the relation between the *t_D_* and *d_max_* of the laser cavitation for water and sea water. Note that data in Figure 12 were cited from the references [33,34] except data of “sea water”. The values “1.49 mg/L” and “9.43 mg/L” revealed the oxygen content of water [33,34]. Figure 12 reveals that the relationship between *t_D_* and *d_max_* is a straight line, even for water with various oxygen contents and temperature. As shown in Figure 12, hemispherical bubbles on the flat specimen were also revealed. As shown in Figure 12, the *d_max_* was proportional to *t_D_*, and the proportional constant of sea water was slightly smaller than that of water. The proportional constant was about 0.0106 for water and 0.0089 for sea water. As such, it can be concluded that the *d_max_* can be estimated by *t_D_*. Thus, in the present paper, *t_D_* was used as a parameter for bubble size.

The *t_D_* in reference [60] was about 750 μs, and the maximum bubble was estimated to be about 7 mm from the relation between *t_D_* and *d_max_* for the sea water, as shown in Figure 12. This corresponded to the longitudinal length of the flattened cavitation that was produced by the pistol shrimp in the reference [60]. As such, *t_D_* of the bubble that was induced by the pistol shrimp can be used for the size of the bubble that was produced by the pistol shrimp.

In the case of Figure 10a, *t_D_* was about 300 μs, the maximum bubble size was about 2.7 mm in diameter, and thus the volume of the bubble was about 10 mm^3^. As mentioned above, the volume of the concavo-convex mating area was about 0.35 mm^3^. Thus, it can be said that the volume of the created cavitation was 30 times larger than the volume of the droplet of the pulsed water jet. Note that the volume of the droplet was assumed from the concavo-convex mating area of the claw.

As *t_D_* can be used for determining the size of the cavitation parameter, as is shown in Figure 12, Figure 13 reveals the relation between *t_D_* and *p_Nmax_*, which was the peak value of the *p_N_* of the pistol shrimp and the pulsed laser. For both cases, coral sand was placed near the bubbles, and the distance between the source of the bubbles and the hydrophone was 90 mm. As shown in Figure 10a, positive peak and negative peak were shown due to the relationship between the directivity of the hydrophone and the position of the shrimp. In the case of the pulsed laser, on the other hand, the hydrophone was set in a position where the noise could be clearly detected. It was reported that the aggressive intensity of the cavitation, i.e., the cavitation erosion rate was estimated from the amplitude of the noise using a pulse height analysis. Then, peak pressure is important to estimate the aggressive intensity of cavitation. The *p_Nmax_* of both cases was roughly proportional to the *t_D_*. When comparing the pistol shrimp and the pulsed laser, the pistol shrimp showed a tendency for the *p_Nmax_* to be higher than that of the pulsed laser when there was an equivalent *t_D_*.

It has been reported that the aggressive intensity of the laser cavitation collapse of degassed water is greater than that of saturated water [33] (which is called the cushion effect). In the case of the pulsed laser, heat was concentrated into the bubble, then the temperature inside the bubble was raised, which caused vapor to form inside the bubble and caused the vapor to act as a cushion effect when the bubble collapsed. Specifically, the *p_Nmax_* of the pulsed laser was slightly lower than that of the pistol shrimp, and this was due to the cushion effect.

### 3.2. Development of Cavitation Generator Mimicking Pistol Shrimp

#### 3.2.1. Cavitation Generator Using Pulsed Laser

Figure 14 reveals the aspect of the cavitation that was induced by the cavitation generator using a pulsed laser, which is shown in Figure 5. As shown in Figure 14, the parallel part of the left-hand side is the nozzle, the diameter of which was 0.5 mm; in addition, the pulsed laser was irradiated in the water-filled chamber, which was placed on the left-hand side of the nozzle. When the pulsed laser was irradiated in the chamber, a laser-induced bubble was generated in the chamber, and the expansion of the bubble also pushed the water. Thus, a pulsed water jet was accelerated through the nozzle. When the pulsed water jet reached the end of the nozzle, a ring vortex cavitation was generated. In the case of a submerged water jet, vortex cavitation of the 0th, 1st, 2nd … order is generated, so from the visualization images, it was inferred the 0th order, i.e., a ring vortex cavitation. This well corresponds to the numerical simulation [77]. Note that the flow field around the pulsed water jet was very important to generate vortex cavitation as shown in reference [77]. Then, the ring vortex cavitation shed from the nozzle to downstream. As shown in Figure 14, when the light source was placed on the other hand of the high-speed video camera, a ring vortex cavitation was observed via a black shape. After *t* = 0.070 ms, the ring vortex cavitation shrunk and then developed again, which then finally collapsed at *t* = 0.200 ms. By measuring the movement of the droplet in the nozzle, the velocity of the pulsed water jet was found to be about 29 m/s. Thus, it can be said that the pulsed water jet was accelerated by the expansion of the laser-induced bubble, which generated the cavitation bubble. The shedding velocity of the ring vortex cavitation was about 20 m/s.

When the pulsed water jet of the droplet, which was 0.5 mm in diameter and 0.5 mm in length, was considered, the energy of the droplet was 1/2 × π × 0.00025^2^ × 0.0005 × 998 [kg/m^3^] × 29^2^ ≈ 166 [mJ] at *v* = 29 m/s. The energy of the used pulsed laser was about 350 mJ. Thus, nearly half of the pulsed laser energy produced the pulse water jet. When the energy of the pulsed water jet that was produced by the claw, which was 0.35 mm^3^ in volume and 29 m/s in velocity, was considered, the energy of the pulsed water jet was 0.16 mJ. Specifically, the pistol shrimp created cavitation bubbles particularly efficiently.

#### 3.2.2. A Cavitation Generator Using a Piezo Actuator

As shown in Figure 14, the submerged pulsed water jet can create a cavitation bubble using a pulsed laser; however, the repetition frequency cannot be increased when compared with a conventional laser cavitation peening system, and this is because the Q-switched Nd:YAG laser is used for the cavitation generator using a pulsed laser. From this, a cavitation generator that uses a piezo actuator was developed.

In the case of the used piezo actuator, when the TTL signal was applied to the actuator, the top of the actuator moved by 69 μm over 0.48 ms. As the diameter of the actuator was 11.5 mm, the movement volume was about 7.16 mm^3^. As such, the velocity of the top of the actuator was 0.14 m/s, and the ideal velocity of the water jet that passed through a nozzle 0.6 mm in diameter was about 53 m/s, as calculated by using the Bernoulli equation.

Figure 15 shows the aspect of a vortex cavitation that was injected by the piezo actuator around the submerged water jet. As shown in Figure 15, the water jet was injected vertically downward by the piezo actuator. As shown in Figure 15, a rig vortex cavitation, whose diameter was about 0.6 mm, was observed as a white one as the light source was placed at the same side of the camera. In order to estimate the shedding speed of the ring vortex cavitation, a yellow dotted line was illustrated near the ring vortex cavitation. When the shedding velocity of the ring vortex cavitation was obtained at *t* = 0.04 ms to 0.16 ms, it was found to be about 5.3 m/s. The aspect of the water jet was determined in order to roughly estimate the jet velocity of the water jet, as shown in Appendix A. The jet velocity was about 5–11 m/s. It was reported that the jet velocity of the high-speed closure of a bioinspired claw reached an asymptotic value in the range of 16–17.5 m/s [77]. In addition, the water jet velocity of a snapping claw was found to be 6.5 ± 1.6 m/s [73], and this might be the velocity of the cavitating region with respect to the schematic drawing in the reference of [73]. As such, it can be said that a developed system that uses a piezo actuator can create a ring vortex cavitation by injecting a pulsed water jet into water. However, the volume of the cavitation was found to be considerably smaller than that of the pistol shrimp. One of the reasons for this was that the water jet velocity was not high enough. In addition, the outlet geometry of the nozzle should also be optimized.

## 4. Conclusions

In order to examine the possibility of developing a novel cavitation generator for mechanical surface treatment, i.e., cavitation peening, a production mechanism for the cavitations caused by a pistol shrimp, i.e., *Alpheus randalli*, was quantitively investigated. As such, two types of cavitation generators were developed. The results obtained can be summarized as follows:Pistol shrimp create a cavitation bubble by producing a pulsed water jet, which is generated by the closing of their claws. With respect to the sample pistol shrimp, the volume of the concavo-convex mating area that was required to generate the pulse jet was about 0.35 mm^3^.The size of the cavitation bubble, which was estimated by monitoring the noise, was about 3 mm in diameter, and the volume was 30 times larger than that of the pulsed water jet.The required speed for the pulsed water jet to create a cavitation bubble was found to be about 10 m/s.The aggressive intensity of the cavitation collapse caused by the pistol shrimp was larger than that of the pulsed laser, even when at an equivalent cavitation volume. This is due to the cushion effect of the pistol shrimp being lesser than that of the pulsed laser.A pulsed laser can generate a pulsed water jet, which can thus create cavitations.The cavitation generator was realized by using a piezo actuator, which was designed to mimic the mechanism of a pistol shrimp. The ring vortex cavitation around the pulsed water jet was also observed.If the cavitation generator using the piezo actuator generate cavitation bubbles that are three mm in diameter, it would be applicable for cavitation peening.

## Figures and Tables

**Figure 1 biomimetics-09-00047-f001:**
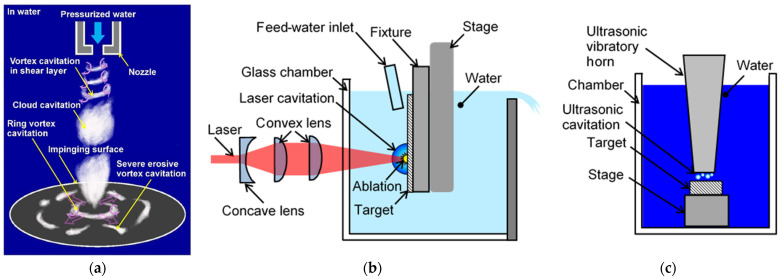
Schematics of cavitation peening system: (**a**) Cavitating jet type [28]; (**b**) Submerged pulsed laser type [33]; (**c**)Vibratory horn type.

**Figure 2 biomimetics-09-00047-f002:**
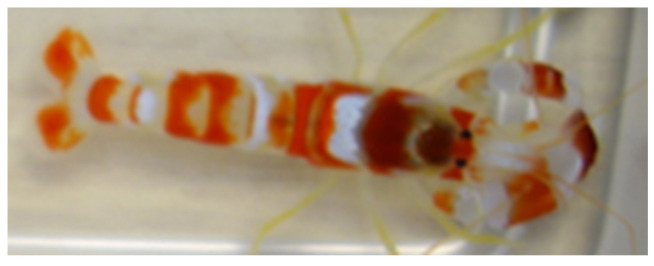
Pistol shrimp (*Alpheus randalli*).

**Figure 3 biomimetics-09-00047-f003:**
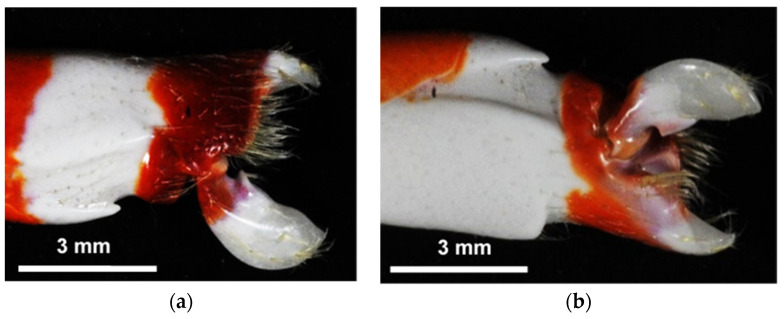
Claw of the pistol shrimp; (**a**) Front side; (**b**) Back side.

**Figure 4 biomimetics-09-00047-f004:**
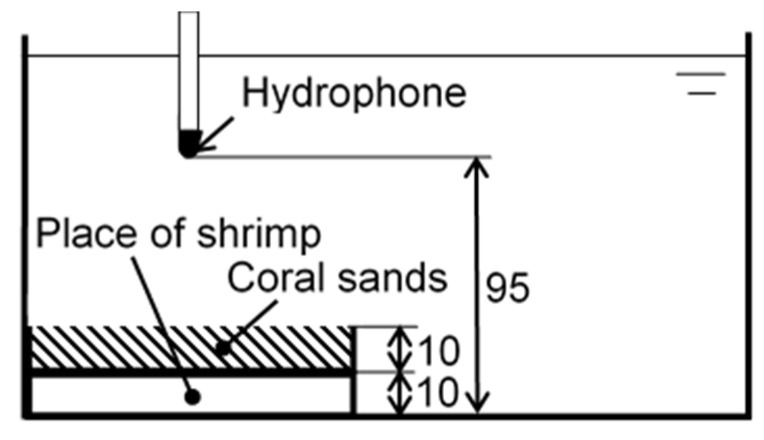
Schematics of the measurement of noise induced by the pistol shrimp (all of the dimensions are in mm).

**Figure 5 biomimetics-09-00047-f005:**
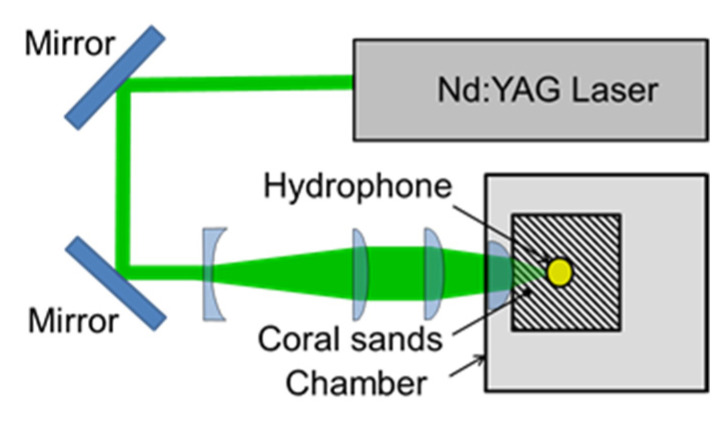
Top view of a pulsed laser system (*λ* = 532 nm).

**Figure 6 biomimetics-09-00047-f006:**
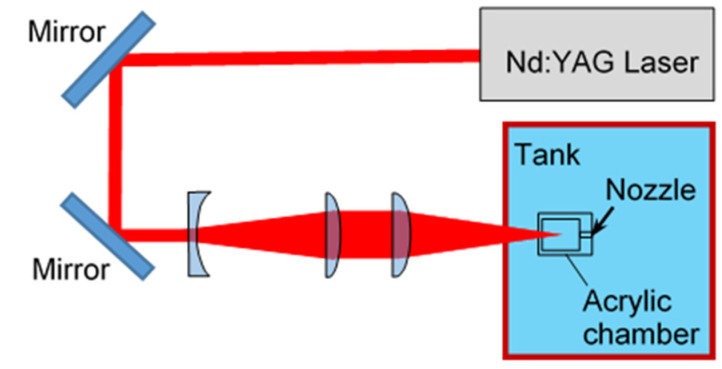
Schematic diagram of a cavitation generator that uses a pulsed laser (*λ* = 1064 nm).

**Figure 7 biomimetics-09-00047-f007:**
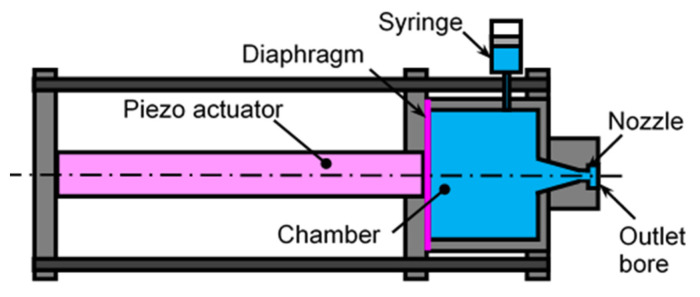
A cavitation generator that uses a piezo actuator.

**Figure 8 biomimetics-09-00047-f008:**
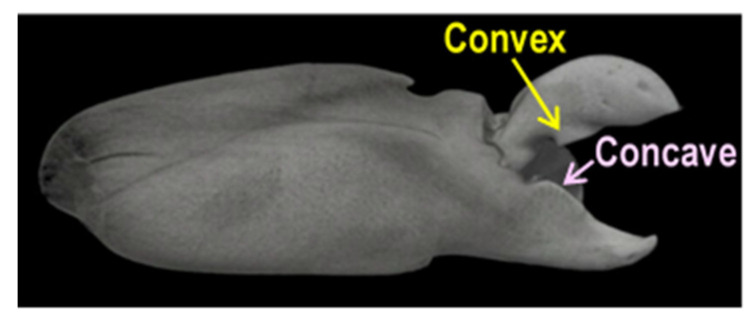
A μCT image of an opened claw.

**Figure 9 biomimetics-09-00047-f009:**
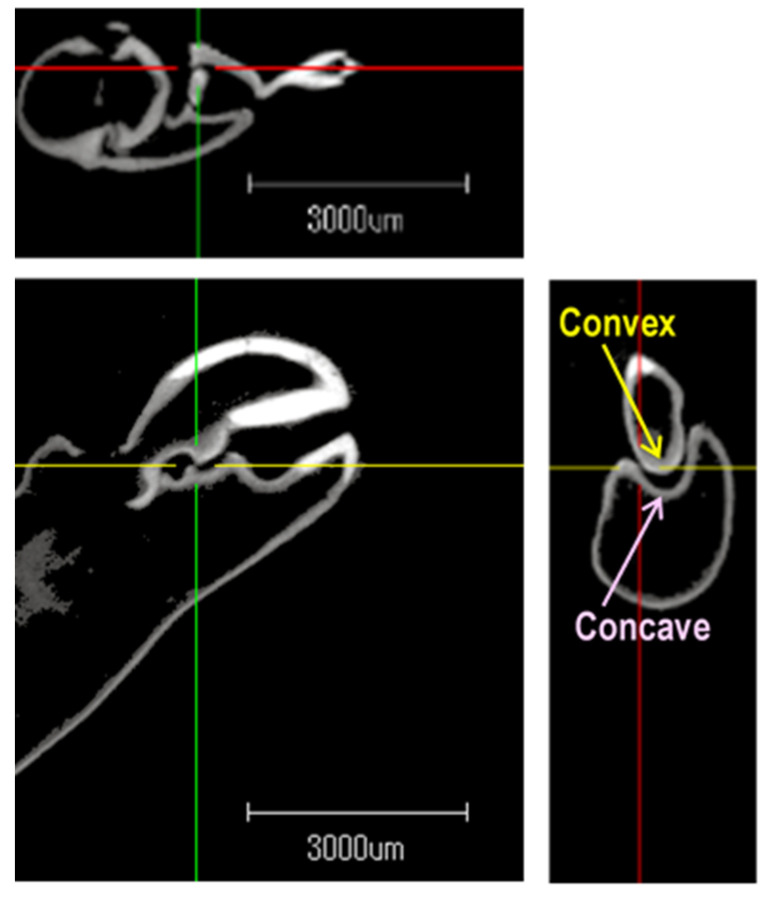
Cross-section image of a closed claw that was observed by μCT.

**Figure 10 biomimetics-09-00047-f010:**
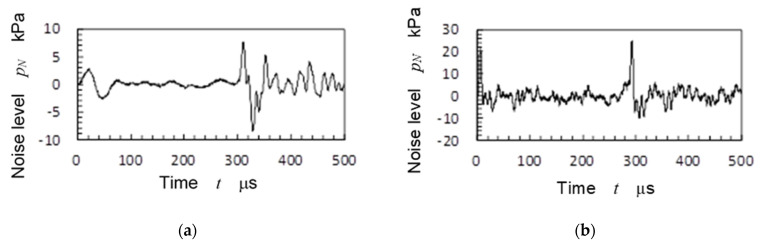
Noise induced by the pistol shrimp and pulsed laser; (**a**) Pistol shrimp; (**b**) Pulsed laser.

**Figure 11 biomimetics-09-00047-f011:**
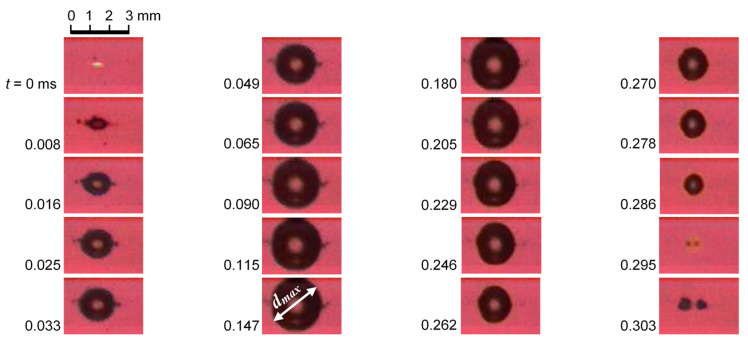
Aspect of bubble induced by pulsed laser observed by the high-speed video camera.

**Figure 12 biomimetics-09-00047-f012:**
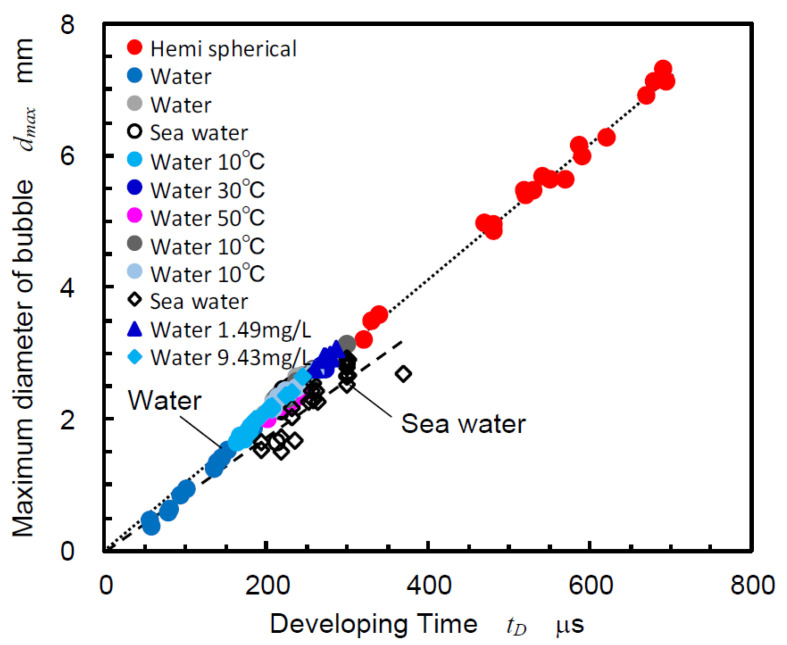
Relation between the developing time and maximum diameter of the laser cavitation. (Data of water was cited from the references [33,34]).

**Figure 13 biomimetics-09-00047-f013:**
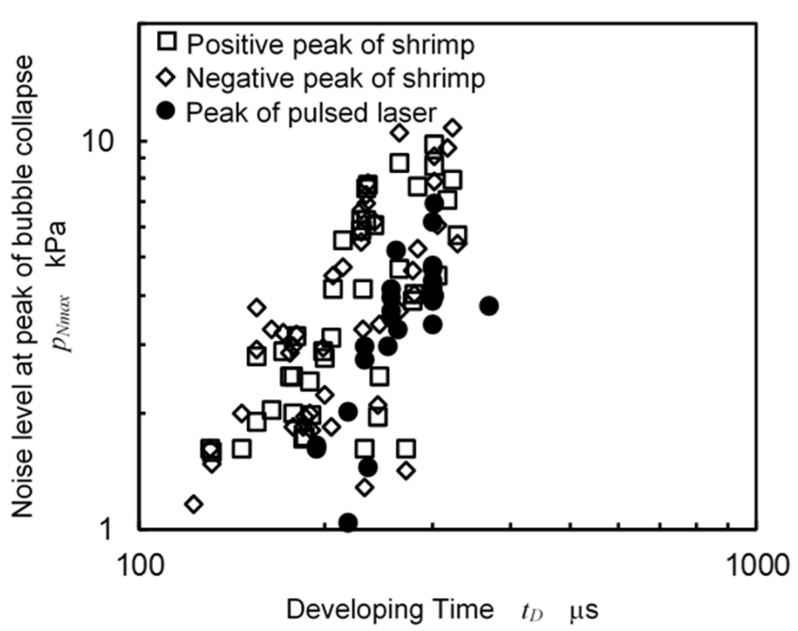
Comparison of noise level at bubble collapse between shrimp and pulsed laser.

**Figure 14 biomimetics-09-00047-f014:**
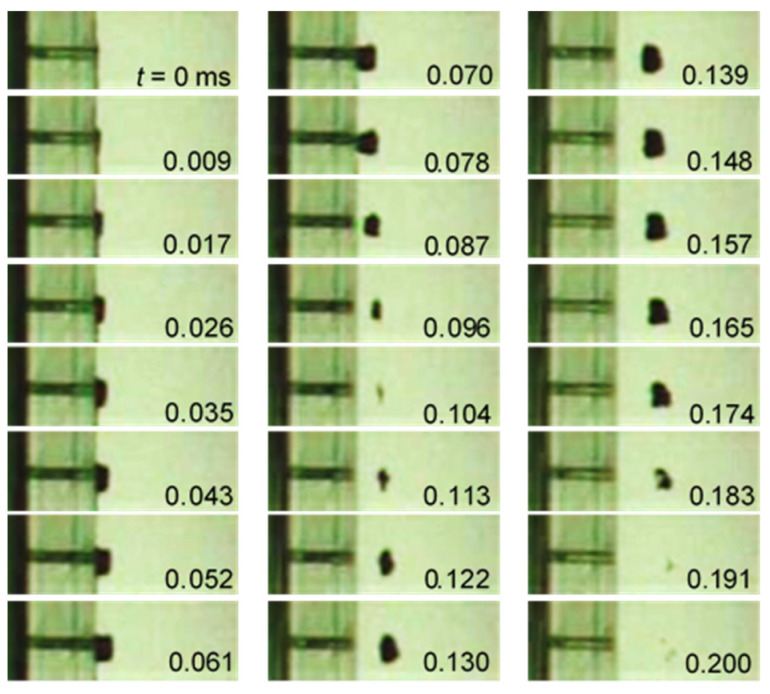
Aspect of the cavitation induced by submerged pulsed water jets using pulsed lasers.

**Figure 15 biomimetics-09-00047-f015:**
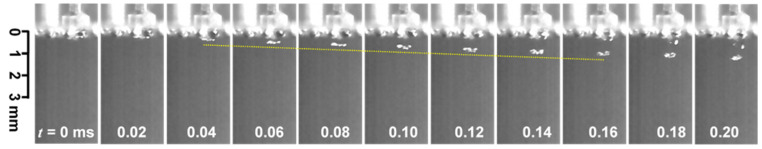
Aspect of the cavitation induced by a submerged pulse water jet using a piezo actuator.

## Data Availability

The data presented in this study are available upon request from the authors.

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
