# Peer review of "Development of a Cavitation Generator Mimicking Pistol Shrimp"

_biomimetics, 2024, doi:10.3390/biomimetics9010047_

Round 1

Reviewer 1 Report

Comments and Suggestions for Authors

I think this paper is interesting and with enough revision it could be published, but some elements of its current state are forcing me to reject it.  Notably the lack of discussion and unacceptable state of Figure 10 which should have been noticed before submission.

The structure of the introduction section jumps around in confusing ways.  The general structure of explaining cavitation, cavitation peening, then cavitation peening methods works well.  What makes it disjointed is the attempts to intersperse motivation in the background discussion, like the final sentence of both the first and second paragraphs that contain statements justifying the study as “worthwhile”.  I would advise moving all discussion of the pistol shrimp to after this initial background on cavitation, and justifying the motivation for the study together at the end of the introduction.

I’m a bit confused by the way conventional vs submerged laser shock peening are separated on page 2.  Both conventional LSP and submerged LSP work via cavitation in a water media, the submergence vs water film is largely irrelevant to the mechanism (other than laser attenuation typically requiring a green wavelength be used instead of IR).  The more typical way LSP methods are broken into categories is whether a sacrificial layer is being used or whether the surface of the material is being directly irradiated, which doesn’t receive any discussion but is generally the difference between the thin water film being used vs full submersion.

Figure 9 has part of a caption cropped that should have been caught if the authors were editing their own paper carefully.  Are there two laser pulses being shown in 9b?  Or is this the initial formation of the bubble registering a sound followed by the cavitation even after 300 us?  What is causing the sound nearest 0 us for the pistol shrimp?  There is practically no scientific or contextual discussion provided by the authors.

Figure 10 is absolutely not acceptable for a scientifically reviewed paper.  How was this data required?  Is this from this paper or multiple references that go without labeling and citation?  What is the difference between the label #2 “water” and #3 “water”.  What is “water 9.43 mg/L”, mg of what?  Hemispherical bubbles on what flat surface?  Once the authors figure out what they’re trying to communicate, they also need find a system so that the labels don’t overlap to the extent some aren’t visible or distinguishable.

In Figure 11 the labeled points are assumedly meant to serve a legend, and should appear as a legend to avoid confusion rather than as three labeled points.  Why are the positive and negative peaks marked for the pistol shrimp but not the pulsed laser?  Assumedly this is because the negative peak is small for the pulsed laser, but why is that?  None of this is discussed at any point.

The fifth conclusion “The energy efficiency to create a cavitation by a pistol shrimp via a pulsed water jet is  about 1,000 times better than that of a pulsed laser” simply isn’t true as it is written.  The energy for the pulsed laser to create cavitation is not the same as the energy required by a pulse laser to create cavitation in the water jet generator shown in Figure 5.  The laser system is creating two cavitations and you’re only measuring the efficiency based on the second one.

What is true, seemingly, is that a piezoelectric system is a more efficient way to power a water jet cavitation device than powering it through laser cavitation in a cavity.  If you want to compare efficiency fairly though you need to compare the energy input vs cavitation energy output of the initial laser cavitation versus your mechanical water jet.  The mechanical system should still be more efficient, but unlikely by the 1000 margin you’re unfairly claiming.

Comments on the Quality of English Language

The quality of writing is intelligible throughout the manuscript, but there are some sections where the grammar is awkward or in some cases confusing or incorrect.  This is understandable as I don’t expect the authors are native speakers, but it will need attention and as a reviewer I’m not getting compensated to make that many line edits.  An example would be page 2 line 63, where it should read “…1/1000-1/100 the value of…”.  Or on page 3 ln 115, it should read “as it is responsible for creating cavitation”.

Reviewer 2 Report

Comments and Suggestions for Authors

The authors present an experimental study of cavitation produced by snapping shrimp and a variety of mimetic devices. The subject should be of interest to the readers of the journal. Much more detail is needed to understand the work that has been performed. Line specific details are listed below.

L18: The abstract should include statements about the study outcomes. How did the devices perform?

L51: How much did the yield stress increase? How big of a benefit may cavitation peening ultimately provide? This is an important part of the motivation for this work.

L58: Please provide a diagram illustrating the descriptions of jet, laser, and horn methods. The text by itself is difficult to understand.

L96: Can the authors state a quantitative goal for how much efficiency gain they want/need to attain?

L110: Can the authors broadly explain why the two generators they developed might be able to meet the efficiency goal that is motivating the work?

L117: Provide details on the uCT (exposure, pixel and field of view size)

L117: The measurement appears to be made above the shrimp. Why was this done? Would the authors expect the sound to have substantial directivity, especially given the water tank boundary conditions? Also, the diagram suggests that the shrimp is below the sand - is this right?

L134: How was the beam diameter determined?

L137: Add a reference supporting the choice of the shorter wavelength.

L144: Provide the frame rate, pixel size, and field of view

L153: Eqn 3 is given in terms of a diameter, while eqn 2 is in terms of radius. Is a factor of two needed here?

L162: For the generator schemes in fig 5 and 6, be very clear about how these are biomimetic, and what the potential advantages are relative to current methods. Also explain what if any simulation or modelling was done to set the design parameters, especially the nozzle dimensions.

L186: Here and throughout this paragraph, explain and label the figures to identify the named features such as deep fossa, large plunger.

L204: Peak pressure is not enough of a descriptor, as the waveform shapes are clearly different. In particular, the peak negative and positive pressures for the shrimp are nearly the same, while for the laser the positive phase dominates. Which of these features is most important for the application? It would also help to look at frequency spectra and time-integrated metrics (energy) of these pulses.

L208: The term 'aggressive intensity' is used several times in the paper but is not defined.

L218: '... the dmax can be estimated by td.' It would be very helpful if the authors could provide a diagram illustrating how this determination is made, starting with a single example of a waveform.

L230: 'the droplet' - please clarify what this refers to

L243: The 'cushion effect' described here makes sense, but it would help to have direct evidence from the video - can the authors directly prove that these bubbles have a slower collapse?

L258: From the images shown, there is no proof that a ring was formed - in this plane we can only see a cloud. Make clear what is measured vs. assumed from geometry.

L270: How much energy went into claw closure? This is the real cost to the animal. Similarly, what is the energy required to make the laser pulse? 

L271. Please label all frames with capture times.

L281: What is 'ideal velocity', and how is it calculated?

L302: The conclusions need to include a discussion of the study limitations and an assessment of whether either of the proposed biomimetic designs can be improved to meet the industrial need (high efficiency peening).

Comments on the Quality of English Language

moderate revisions are needed. Please note the spelling of 'mimicking'

Reviewer 3 Report

Comments and Suggestions for Authors

I found this manuscript very interesting.

In the present paper, in the first part, a pistol shrimp claw, i.e., alpheus randalli, was measured by a mCT. It would be a palatable addition to the reader if the authors state explicitly in their manuscript how this was done.

Additionally it would be very much appreciated if the authors stated explicitly in their manuscript how was the volume of the pulsed water jet estimated.

The authors state in their manuscript that:

'Then, a cavitation bubble induced by a pistol shrimp was evaluated by comparing the noise created by laser cavitation. In the second part, a pulsed water jet is produced by mimicing a  pistol shrimp, and two cavitation generators are developed using a pulsed laser and a 110 piezo actuator.'

It would be a significant addition to the quality of their manuscript if they stated how was this achieved and point this out in their conclusions. This should be coupled with real time indication of the creation of the cavitation induced by a submerged pulse water jet. Did this action not influence the surrounding fluid dynamics? I mean was this action non-intrusive?

Can the authors point out how point 4. of their conclusions can be improved?Why are the authors convinced that it can be said that a developed system that uses a piezo actuator can create a ring vortex cavitation by injecting a pulsed water jet in water? They point this out at the beginning of page 10 of their manuscript.

I noticed that he authors have some indication of real time performance of their system in figure A1. Whys is the Appendix needed? Why can the Appendix not be absorbed within the manuscript? Is it different from the rest of the main manuscript and if yes in what respect?

Otherwise I think that the manuscript can proceed to publication.

Comments on the Quality of English Language

I find this an interesting paper. It is a well written and carefully prepared manuscript.

Round 2

Reviewer 1 Report

Comments and Suggestions for Authors

The authors were able to address my major concerns of scientific nature and improve the discussion, I believe several of my early comments arose from confusions that are less likely to occur with the newer text.  The author's separation of submerged vs. film shrouded laser peening makes sense supplying the film is thin enough the cavitation bubble can't form (though I've often seen the flowing water supplied to be thick enough they can).

I'm still a bit disappointed with the state this manuscript was initially submitted in, as the inattention to detail doesn't provide a great deal of confidence in the reliability of the experiments.  Figure 12 (previously Figure 10) has been improved, though each data series should still be linked to the reference it came from as is typical in many journals.  More pressingly the legend in Figure 13 seems incomplete and needs to be fixed. If careless mistakes like these are present in the final submission it calls into question if the underlying data has similar errors.

From a discussion standpoint I still think this paper is disjointed. For example the sentence about magnesium in the abstract has almost no bearing on the subject matter of the paper.  In a high impact journal this would typically be grounds for rejection and resubmission, but the editors of the journal appear to want to publish it so I question whether anything I write would change that and I'll leave that decision up to them.

Comments on the Quality of English Language

The quality of the English is readable, though the writing remains grammatically stilted in places.  There are a handful of actual minor errors remaining, for example pg2 ln49 should be "increasingly high strain rates", but they should be caught easily in standard final copyediting.

Reviewer 2 Report

Comments and Suggestions for Authors

The authors have improved the manuscript and provided most of the requested details. 

Comments on the Quality of English Language

Please run another grammar and spelling check (e.g. L20 'gnateaters').
